# Structural basis of a redox-dependent conformational switch that regulates the stress kinase p38α

Joan Pous [1,7], Blazej Baginski [1,5,7], Pau Martin-Malpartida [1,7], Lorena González[1,6], Margherita Scarpa[1], Eric Aragon[1], Lidia Ruiz[1], Rebeca A. Mees [1], Javier Iglesias-Fernández[2], Modesto Orozco [1,3], Angel R. Nebreda [1,4] ✉ & Maria J. Macias [1,4] ✉

Many functional aspects of the protein kinase p38α have been illustrated by more than three hundred structures determined in the presence of reducing agents. These structures correspond to free forms and complexes with activators, substrates, and inhibitors. Here we report the conformation of an oxidized state with an intramolecular disulfide bond between Cys119 and Cys162 that is conserved in vertebrates. The structure of the oxidized state does not affect the conformation of the catalytic site, but alters the docking groove by partially unwinding and displacing the short αD helix due to the movement of Cys119 towards Cys162. The transition between oxidized and reduced conformations provides a mechanism for fine-tuning p38α activity as a function of redox conditions, beyond its activation loop phosphorylation. Moreover, the conformational equilibrium between these redox forms reveals an unexplored cleft for p38α inhibitor design that we describe in detail.

Protein kinases play a critical role in most signal transduction pathways, transmitting signals from the plasma membrane to the interior of the cell. These signals result from ligand-receptor interactions as well as from environmental perturbations, and modulate cell proliferation and differentiation or regulate immune responses and many other specialized cell functions[1]. p38α (MAPK14) is a member of the mitogen-activated protein kinase (MAPK) family that phosphorylates target proteins on serine and threonine residues usually followed by a proline. The kinase activity of p38α plays an active role in the regulation of gene expression programs and other cellular processes that control the differentiation and survival of many cell types. The p38α pathway responds to numerous stress signals, including oxidative stress, which is caused by an imbalance between the levels of oxidants and the cell's ability to execute an effective antioxidant response[2,3].

Reactive oxygen species (ROS), like hydrogen peroxide ($H_2O_2$), are important modulators of signaling pathways, for example through redox modification of Cys residues[4–8]. There is evidence indicating that ROS triggers p38α activation indirectly through the upstream kinases ASK1 (MAP3K7) and MTK1 (MAP3K4). ASK1 activation involves Cys oxidation of the ASK1 inhibitory protein Thioredoxin, which triggers its dissociation from ASK1, and MTK1 activation involves coupled oxidation-reduction modifications of specific Cys residues[9,10]. On the other hand, Cys oxidation of the MAP2K MKK6 (MAP2K6) prevents the phosphorylation of p38α and its subsequent activation[11]. These observations indicate that ROS can potentially modulate the activity of different p38α pathway regulators with opposite outcomes.

The p38α sequence contains three conserved Cys residues and one vertebrate-specific Cys residue (Cys119) (Fig. 1a, b). Cys 39 and Cys

[1]Institute for Research in Biomedicine (IRB Barcelona), The Barcelona Institute of Science and Technology, Baldiri Reixac, 10, 08028 Barcelona, Spain. [2]Nostrum Biodiscovery, Josep Tarradellas 8-10, 3-2, 08029 Barcelona, Spain. [3]Departament de Bioquímica i Biomedicina, Facultat de Biologia, Universitat de Barcelona, 08028 Barcelona, Spain. [4]Institució Catalana de Recerca i Estudis Avançats (ICREA), Passeig Lluís Companys 23, 08010 Barcelona, Spain. [5]Present address: Global Health Medicines R&D, GSK, c/ Severo Ochoa, 2, 28760 Tres Cantos, Madrid, Spain. [6]Present address: Grupo Menarini España, c/ d'Alfons XII, 587, 08918 Badalona, Barcelona, Spain. [7]These authors contributed equally: Joan Pous, Blazej Baginski, Pau Martin-Malpartida. ✉e-mail: angel.nebreda@irbbarcelona.org; maria.macias@irbbarcelona.org

221 are partially buried in the structure and are redox-insensitive. In contrast, Cys119 and Cys162 are solvent-exposed and have been shown to be sensitive to redox changes in biochemical assays, yielding an inactive kinase despite being phosphorylated in cells treated with either H$_2$O$_2$ or prostaglandin J2[13]. Cys119 also participates in the interaction with the MAP2K MKK3b (MAP2K3 isoform b)[14]. In fact, whether p38α samples conformations of an oxidized state remains unknown because all p38α structures, both in solution and crystallized, have been determined either in the presence of reducing agents or after mutation of Cys119 to Ser to avoid Cys-induced protein aggregation[15,16].

In this work, we aim to investigate how p38α responds to oxidative changes in cells by studying the structure of the purified protein previously exposed to air oxidation overnight. Under these non-reducing conditions, we obtained diffraction quality crystals showing that Cys119 and Cys162 can form an intramolecular disulfide bond that induces the partial unwinding of the αD helix, which in turn alters the shape and charge distribution of the kinase docking site. As a consequence of this change, the oxidized form compromises the interaction of p38α with modulators and substrates, including allosteric activators such as the transforming growth factor-β-activated kinase 1 (TAK1)-binding protein 1 (TAB1) that we have experimentally analyzed. Furthermore, the redox-dependent conformational equilibrium of the αD helix revealed the presence of an unexplored pocket for drug discovery. Molecules that bind to the cleft present in the oxidized state are predicted to compromise interactions with activators and substrates, even in the absence of the disulfide bond, opening up opportunities for the development of specific p38α inhibitors.

## Results

### p38α forms an intramolecular disulfide bond between Cys119 and Cys162

Cys162 is located in the ED loop, which participates in substrate docking, whereas Cys119 is at the end of the αD helix (formed by residues 113–119), and it is close to the docking groove (Fig. 1b). We noticed that the αD helix and its surrounding loops are indeed partially disordered in about a quarter of the p38α structures determined so far (84 out of 360), with Cys119 occupying different orientations in the crystals (Fig. 1b, Table 1). In these structures, the distances between the thiol groups of Cys119 and Cys162 range from 8 to 19 Å indicating an important degree of dynamic behavior in this area (Fig. 1c). We hypothesized that this flexibility might allow the sulfhydryl groups of Cys119 and Cys162 to come close enough in space to form an intramolecular disulfide bond in response to redox changes inside the cell. We purified the p38α recombinant protein using 1 mM DTT, and left the diluted protein solution (60 μM) dialyzing to buffer without DTT at 4 °C under air oxidation overnight. We confirmed that under these conditions the protein did not aggregate and showed cooperative unfolding transition using Differential Scanning Fluorimetry (DSF), with similar stability as the reduced protein purified under high DTT concentrations (Fig. 1d). Given that the four Cys residues present in p38α are located outside the ATP binding site (Fig. 1b), we used the pyridinyl imidazole SB203580 compound[17], a p38α ATP competitor for crystallization.

We obtained crystals in several conditions, with the two best diffracting ones being solved by molecular replacement using the PDB: 4LOO structure as a search model. Images of the asymmetric unit and snapshots of the density maps are shown in Supplementary Fig. 1. The statistics for data collection and structure refinement are summarized in Table 2. During the refinement process of our data, we confirmed the presence of the characteristic N and C lobes present in all MAPK structures[15,18,19] and observed the presence of a spontaneously formed intramolecular disulfide bond between Cys119 and Cys162 (Fig. 2a–c). The other two Cys residues present in p38α remained reduced. The two oxidized structures superimposed closely, with a Root Mean

Square Deviation (RMSD) of 0.224 Å (Fig. 2a) and relatively well to that of the reduced form of p38α (RMSD 0.90 Å) (Fig. 2d). In these p38α structures, the SB203580 molecule was located in the hinge region (catalytic site) between the N and C lobes, binding in a DFG-in conformation. The molecule occupied the same position as previously observed in other reduced p38α structures, confirming that the ATP-binding site is not affected by the presence of the disulfide bond (Supplementary Fig. 1b). In the oxidized form, as well as in 264 out of the 360 p38α structures deposited in the PDB, the activation loop (A-loop) was flexible. This flexibility was deduced from the moderately visible electron density signal that can be only fully traced for the starting and end regions, including the Asp-Phe-Gly motif (DFG, residues 168–170) and the Ala-Pro-Glu motif (APE, residues 190–192), respectively.

### Differences between reduced and oxidized forms of p38α

The most striking feature of the oxidized form was that the disulfide bond induced a substantial conformational rearrangement in the regions where the two Cys residues were present. We observed the absence of the αD helix in the oxidized form as compared with the reduced one (Supplementary Fig. 2). In addition, Cys119 showed a significant shift of ~6.7 Å from the position it occupies in reduced structures and now lies adjacent to the ED loop, a short motif that contributes to p38α target binding and specificity, and where Cys162 is located (Fig. 2b–d).

### Oxidized p38α has a binding cleft with potential pharmacological applications

Remarkably, comparison of the oxidized form with the known reduced structures of p38α revealed differences affecting the morphology of the docking groove and nearby regions directly affected by the formation of the disulfide bond. Simultaneously with the formation of the bond, the short αD helix unwinds and adopts an extended conformation, which we refer to as loop αD (L$_D$) (Fig. 2d). The presence of the disulfide bond also changed the shape, hydrophobicity and electrostatic properties of the protein's surface with respect to the reduced conformations, particularly in the docking site region, thereby highlighting the potential impact of the oxidized conformation on the p38α functions (Fig. 3a, b) (Supplementary Movie 1).

We observed that the disulfide bond closes the canonical docking groove and creates a cleft in the area previously occupied by the αD helix. This cleft (highlighted in yellow, Fig. 3a) was identified using the SiteMap tool (Schrödinger package. SiteMap Score = 0.58), which facilitates the identification and analysis of binding sites and predicts the druggability of targets. The cleft is slightly larger than the canonical docking groove, as reflected by the solvent-accessible surface area values (SASA = 182 Å$^2$ and 157 Å$^2$, respectively) calculated using the fpocket software[20]. The inner part of the cleft is mostly composed of hydrophobic and aromatic residues, surrounded by polar and charged amino acids exposed to the surface. The specific residues defining the cleft are shown in Fig. 3c and Supplementary Fig. 3, side by side with those of the canonical ED binding site. Although some residues are conserved between the canonical and the cleft, their positions in the cavities are different (residues highlighted in bold in Fig. 3c).

Notably, the disulfide bond is also expected to form in p38β, which has these two Cys residues conserved, but not in the p38γ and p38δ family members or in other MAPKs such as JNKs and ERKs (Fig. 1a), opening up new possibilities for the design of pharmacologically active compounds specifically targeting p38α and p38β.

### Interconversion of oxidized and reduced conformations

To characterize the reversible interconversion of the oxidized and reduced forms of p38α, we performed molecular dynamics (MD) simulations starting with the oxidized state (PDB:8ACM), with Cys119 and Cys162 in the thiol state but as close to each other as ~3 Å distance.

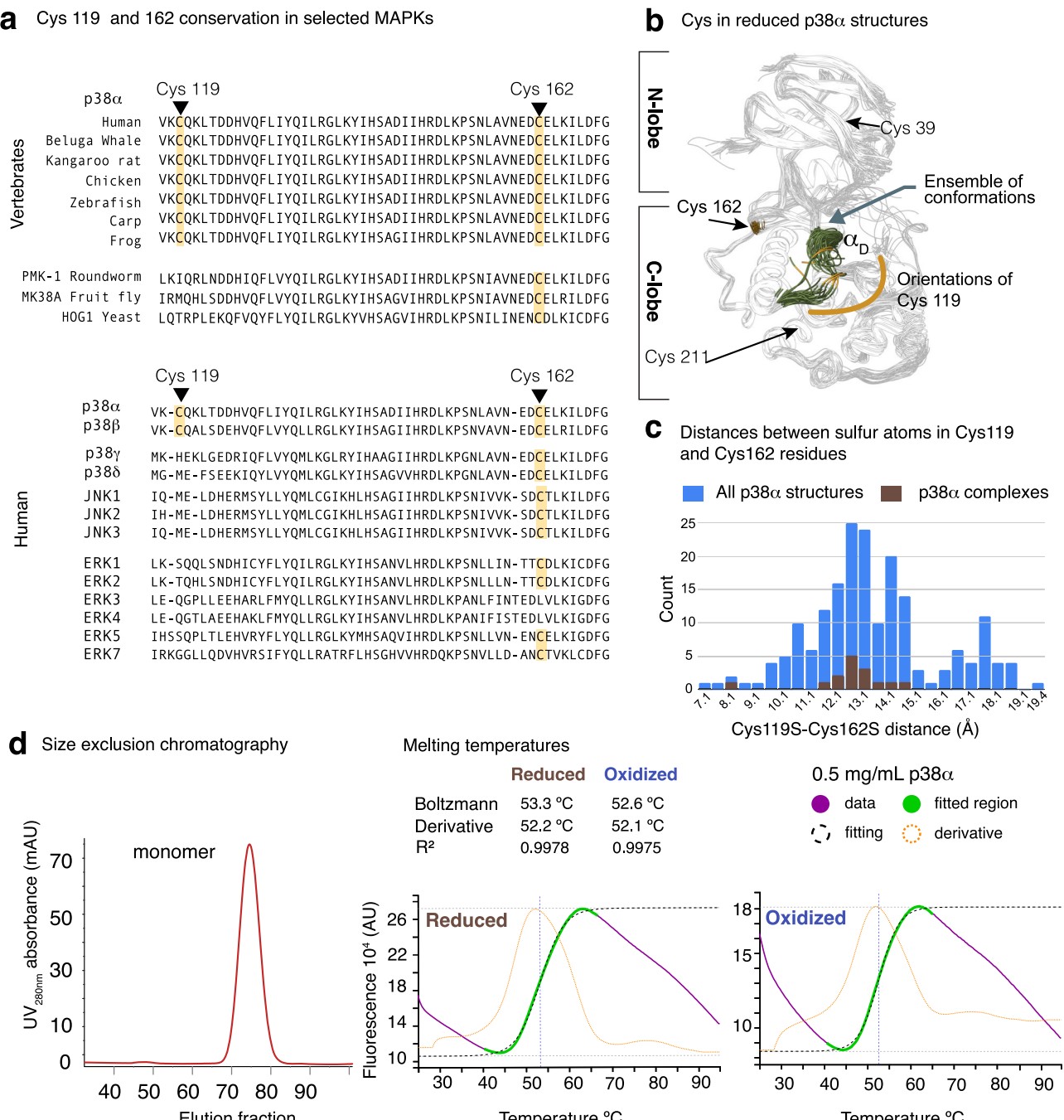

**Fig. 1 | Conservation and 3D distribution of the Cys119 and Cys162 pair, and biophysical characterization of recombinant p38α. a** Top: Sequence alignment of the region containing Cys119 and Cys162 in p38α showing the high conservation of the two Cys residues in vertebrates. Bottom: Sequence alignment of the region containing Cys119 and Cys162 in p38 family members and other human MAPKs, showing that Cys119 and Cys162 are only conserved in p38α and p38β. Cys positions are marked yellow. **b** Reduced p38α structure, shown as ribbon and overlaid with a transparent surface. Superposition of 60 p38α reduced crystal structures (colored in light gray) displaying the high overall structural similarity and the conformational sampling of the αD helix (in green). The N and C terminal lobes of the kinase and the position of the four Cys residues are labeled. The side chains of the solvent exposed Cys residues (119 and 162) are shown. The variable orientations sampled by Cys119 in the ensemble are indicated as an orange semicircle. PDB entries used for the figure are collected in Table 1. **c** Histogram showing the distances between the sulfur atoms of Cys119 and Cys162 in all structures deposited in the PDB (blue bars). Complexes with a peptide ligand bound to the docking groove of p38α are shown in brown bars to emphasize that these complexes select Cys separated by distances in the range of 13 Å, consistent with reduced structures. **d** Left: Purity and monomeric state of oxidized p38α shown by size exclusion chromatography. Right: Protein thermal shift comparison between the reduced and oxidized forms of p38α (0.5 mg/mL) using SYPRO Orange Protein Gel Stain and analyzed using the HTS explorer[12]. Experiments were performed in duplicates. Source data for this figure are provided as a Source Data file.

**Table 1 | PDB structures used to generate Fig. 1b**

| PDB entries |
| --- |
| 1KV1, 1KV2, 1OZ1, 1YWR, 2BAJ, 2BAL, 2FSL, 2FSM, 2FSO, 2FST, 2EWA, 2GHM, 2NPQ, 2YIS, 2YIW, 2YIX, 3DS6, 3FI4, 3FL4, 3HV7, 3K3I, 3K3J, 3L8S, 3MW1, 3OEF, 3S4Q, 3ZSI, 4AAO, 4AAC, 4DLI, 4E5A, 4E5B, 4E8A, 4EH5, 4EH9, 4GEO, 4KIP, 4L8M, 5N63, 5N64, 5N67, 5N68, 5O8V, 5OMG, 5OMH, 5R8W, 5R9K, 5RA8, 6HWU, 69ML, 6QDZ, 6SFO, 6ZWP, 7BDO, 7BE4. |

**Table 2 | Crystallization, data collection and refinement statistics**

| PDB ID | 8ACO | 8ACM |
| --- | --- | --- |
| Content | Oxidized p38α + SB203580 | Oxidized p38α + SB203580 |
| Crystallization | | |
| Conditions | 0.1 M Magnesium chloride hexahydrate. 1 M HEPES pH 7.0 15% w/v PEG 4000 (PROPLEX B12) | 0.1 M HEPES pH 7.5 25% w/v PEG 3350 (TOP 96 C7) |
| Temperature | 4 °C | 4 °C |
| Cryo protection | 10% glycerol | 10% glycerol |
| Data collection | | |
| Beamline data | XALOC-BL13 (Feb. 2021, Alba, Barcelona) Detector: Pilatus 6 M, Wavelength: 0.9792 Å, Temperature: 100 K | |
| Distance | 463.1 mm | 381.9 mm |
| Data processing | | |
| Mosaicity | 0.85° | 0.46° |
| Space group | P $2_1 2_1 2_1$ | P $2_1 2_1 2_1$ |
| Cell | 65.30,74.61,78.10 Å$^3$ | 65.32,75.16,78.67 Å$^3$ |
| Resolution range | 78.10 – 2.65 Å (2.79-2.65 Å) | 78.67 – 2.14 Å (2.26-2.14 Å) |
| Multiplicity | 5.6 (5.7) | 5.5 (5.4) |
| Completeness | 94.1% (84.0%) | 94.3% (93.9%) |
| I/σ(I) | 13.6 (4.4) | 10.6 (3.2) |
| Rmerge | 0.075 (0.397) | 0.085 (0.380) |
| Wilson B factor | 44 Å$^2$ | 33 Å$^2$ |
| Structure determination and refinement | | |
| Method | Molecular Replacement from PDB model 4LOO | |
| R factor | 0.21 (0.38) | 0.20 (0.24) |
| Free R factor | 0.28 (0.55) | 0.24 (0.26) |
| RMS bonds | 0.002 Å | 0.005 Å |
| RMS angles | 1.21° | 1.29° |
| RMS chiral | 0.04 Å$^3$ | 0.06 Å$^3$ |
| <B> | 58 Å$^2$ | 46 Å$^2$ |
| Ramachandran: Favored, Outliers | 97%, 3% | 95%, 5% |

Highest resolution shell shown in parentheses.
RMSD between both structures 0.224 Å (2193 atoms).

During the simulations, we observed that the position of the two Cys residues evolved in the direction of the open conformations. Histogram analysis of the distances between Cys119 and Cys162 along the MD simulations (Fig. 4a) showed that the majority are in the range of 10-13 Å, compatible with the reduced structures, recapitulating the observations of the structures available in the PDB as reported in Fig. 1c.

Similarly, a Gaussian accelerated MD (GaMD) simulation starting from the fully reduced form of p38α also evolved to conformations with both Cys residues at distances as close as 3.1 Å. These distances are consistent with a disulfide bond forming under appropriate redox conditions (Fig. 4b), thus supporting the view that both reduced and oxidized forms can interconvert. Supplementary Movie 1 illustrates the conformational changes between the oxidized and reduced p38α

forms. Snapshots of the GaMD simulation are shown (Fig. 4b) to emphasize that the relocation of Cys119 to get close to Cys162 also affects the secondary structure of the neighboring residues, as observed in the crystals of the oxidized form. The evolution of the p38α structure during the two MD simulations is shown in Supplementary Fig. 4a, b.

## Functional implications of the redox-modulated p38α conformational changes

The disulfide bond closes the docking groove required for the binding of p38α to activators such as MKK3b (Supplementary Movie 2) and MKK6, and to phosphatases like MKP5, but also to substrates such as MEF2A and MK2[14,21,22]. In fact, modeling the structure of the corresponding complexes with the oxidized p38α form suggests that its interaction with the targets is hampered (Fig. 5a, Supplementary Fig. 5a). In the case of MKK3b, the interaction is even more disfavored, since its binding is facilitated by intermolecular contacts between Cys119 in p38α and the Cys in position φB + 2 of the ligand, and these contacts will be precluded by the intramolecular disulfide bond in the oxidized form of p38α. The structural modifications will also likely interfere with the binding to TAB1, which promotes p38α autophosphorylation[23] thus affecting the non-canonical activation pathway. As indicated in Fig. 5a, the oxidized conformation will disturb the correct positioning of the C-terminal region of TAB1 (residues 403–411) at the canonical p38α binding site. The expected effects of the redox changes in p38α on the binding of TAB1 are illustrated in the Supplementary Movie 3.

To provide experimental evidence for the impact of the intramolecular disulfide bond formation on p38α binding capacities, we performed electrophoretic mobility shift assays (EMSA) using a TAB1 peptide that has been previously reported to bind to p38α and induce its activation[23]. We observed that whereas the reduced p38α clearly interacted with the TAB1 peptide, the oxidized protein was not able to shift the electrophoretic mobility of the peptide at the same range of concentrations (Fig. 5b, Supplementary Fig. 5b). We quantified the affinity of the TAB1 peptide for either the reduced or the oxidized forms of p38α using isothermal calorimetry (ITC). As in the EMSA, the reduced form of p38α was able to interact with TAB1 in the low micromolar range, but no binding was detected with the oxidized protein (Fig. 5c). Remarkably, the binding capacity was recovered after dialyzing the oxidized p38α protein in the presence of the reducing agent tris(2-carboxyethyl)phosphine (TCEP) (Fig. 5d).

All in all, our results indicate that the oxidized p38α structure has an accessible ATP-binding region and a flexible A-loop, which are compatible with its activation. However, the formation of the disulfide bond induces a pronounced conformational change in the binding groove, a region used for the interaction of p38α with both activators and substrates. As a consequence, this conformational switch prevents activators and substrates from binding efficiently, affecting the function of p38α.

## Discussion

Protein oxidation and phosphorylation-based signaling networks cooperate to regulate how cells respond to different amounts of oxidants during homeostasis and under oxidative stress[4]. In fact, there is evidence that oxidation can affect the activity of some protein

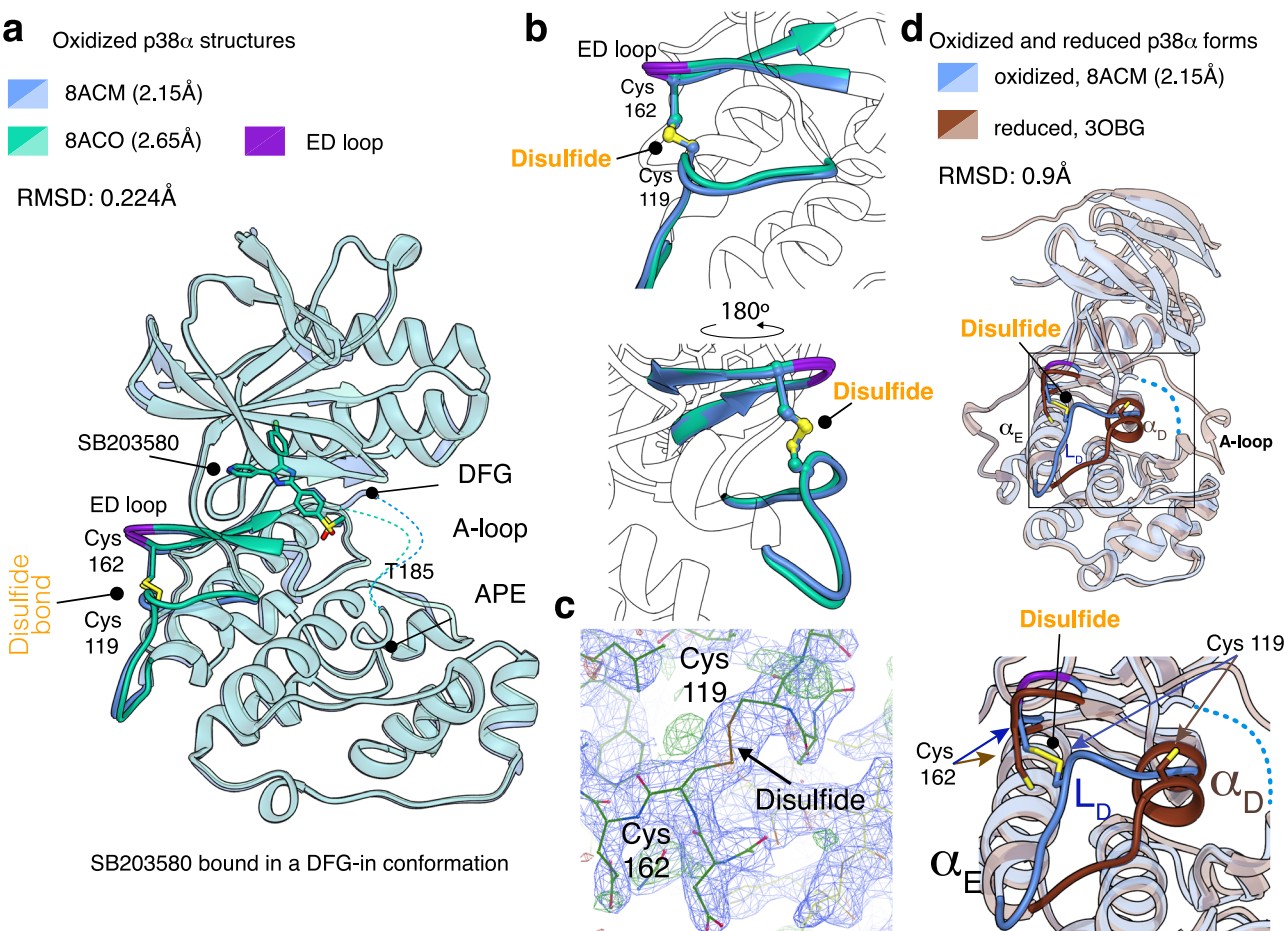

**Fig. 2 | Structures of the oxidized p38α. a** Superposition of the two oxidized structures (light blue and green, respectively). The localization of the disulfide bond (yellow) is indicated. The SB203580 molecule used for crystallization is bound in a DFG-in conformation. In both structures, the A-loop is flexible and is represented by dotted lines. **b** Close up of surrounding regions to the disulfide bond, which are colored in blue and green. The two views are rotated by 180°. The ED loop is colored in violet whereas the disulfide bond is shown using a ball and stick representation and indicated with a rounded arrowhead. **c** Electron density map plotted at 1σ. The final refined structure is fitted to the density and shown in green for clarity. The electronic density directly after molecular replacement is shown in Supplementary Fig. 2. The disulfide bond connecting Cys119 and Cys162 is indicated by an arrow. **d** Structural comparison of the reduced and oxidized forms of p38α, indicating the differences in $L_D/\alpha_D$ region, with a close up of this area shown in the bottom panel. Supplementary Movie 1 illustrates the conformational changes between the oxidized and reduced p38α forms.

kinases[7,13]. For instance, the p38 MAPK activator MKK6 is inactivated by oxidative conditions, which favor the formation of an intramolecular disulfide bond that inhibits ATP binding[11]. This mechanism could extend to other MAP2K family members, since the two Cys involved are conserved. Another example of redox control via disulfide bond formation is the inactivation of the serine/threonine-protein kinase Akt2, which is involved in insulin signaling[24]. Whereas in the case of MKK6 the structural details have not been explored to date, the structure of Akt2 shows the presence of an intramolecular disulfide bridge that induces the unfolding of most of its αC helix, occluding the ATP binding cavity and interfering with the catalytic capacity. Interestingly, we found that the disulfide bond we have identified in p38α has a different role from the above kinases, because it affects neither ATP binding nor the flexibility of the A-loop. In the case of p38α, the disulfide bond changes the conformational properties of the binding groove, preventing its interaction with activators and substrates. This oxidized state could have a protective role during cellular homeostasis, ensuring that p38α signaling is not erroneously activated in response to low levels of oxidants[8,25,26]. Likewise, during the early stages of oxidative stress, antioxidant molecules such as thioredoxins and glutathione S-transferases, can trigger p38α reduction. In this manner, the concentration of activatable p38α could be modulated,

either by upstream MAP3Ks, such as MTK1, which through the phosphorylation of MAP2Ks induce the canonical pathway[9,10] or through the non-canonical activation mediated by TAB1[23]. In this context, the antioxidant machinery would facilitate both the reduction and phosphorylation of p38α, thereby ensuring the orchestration of the appropriate cellular responses. However, if the ROS concentrations are too high, cells may not be able to yield sufficient levels of reduced and activated p38α, resulting in the formation of a disulfide bond that decreases the capacity of p38α to interact with and phosphorylate its targets. This conformational change provides a mechanistic explanation for a previous report describing that treatment of cells with high concentrations of $H_2O_2$ increased the activation loop phosphorylation of p38α but inhibited its kinase activity[13]. Based on all these observations, we propose that both the phosphorylation of the activation loop and the redox state of p38α should be considered as markers of the active or inactive state of this kinase (Fig. 5e).

The p38α signaling pathway has been associated with several human pathologies, such as chronic inflammation, cancer, and cardiovascular and neurodegenerative diseases, which has attracted the attention of pharmaceutical companies and academic laboratories worldwide [2,27,28]. However, research endeavors have focused on designing inhibitors for the reduced conformations and the

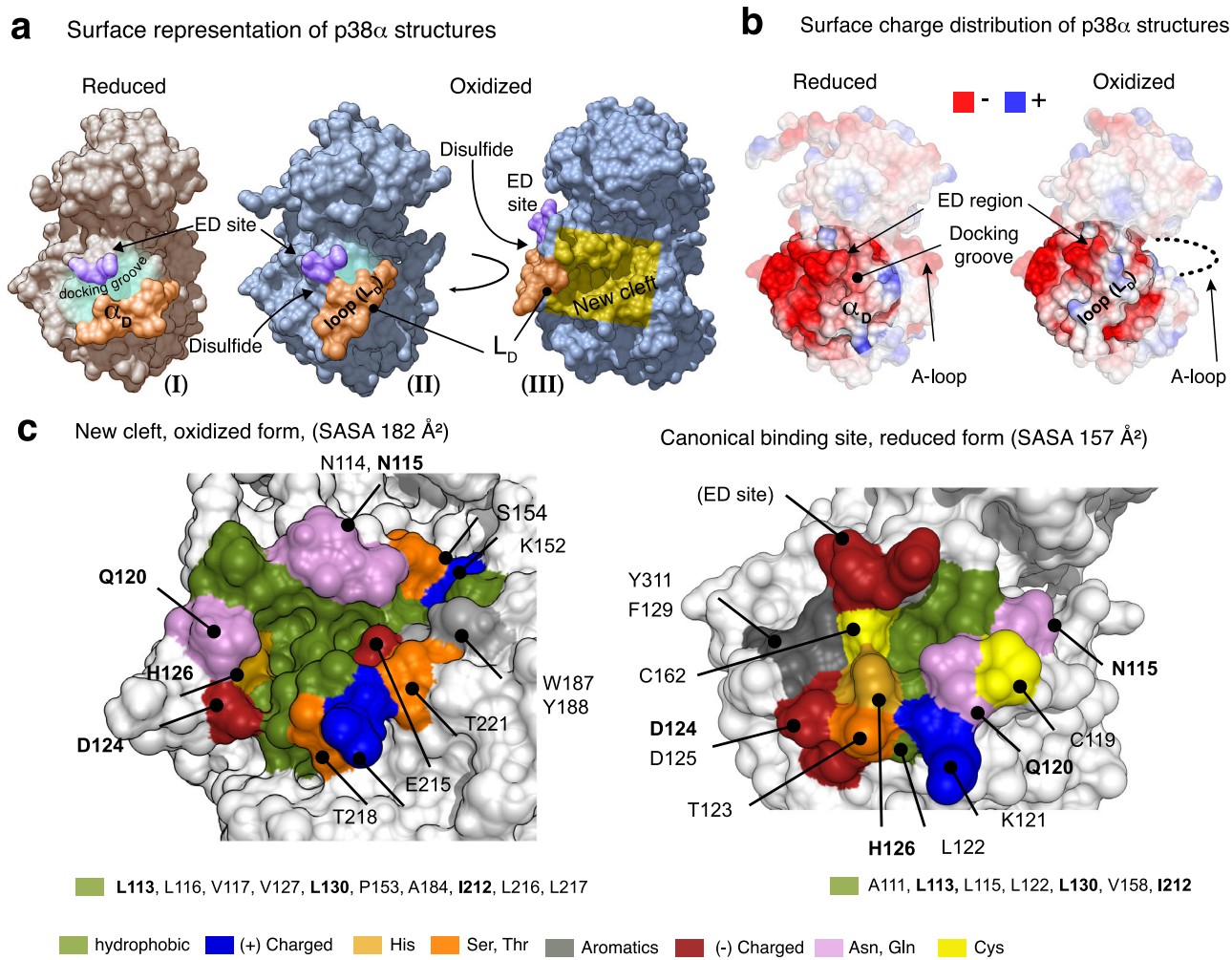

**Fig. 3 | Structural properties of the canonical binding site and the cleft of p38α.**
**a** Surface representation of the reduced (I, beige) and oxidized (II, and III, blue) structures, with the location of the ED site, disulfide bond, and the αD helix/LD loop indicated. In III, the disulfide bond is at the back (indicated by an arrow). In the oxidized form, the LD loop is shifted by ~45° with respect to the original αD helix observed in the reduced structure, compressing the original docking groove. At the same time, a cavity is created where the αD helix was previously located. In order to observe this cleft, the protein surface III is rotated by 90° with respect to the view shown in II. The cleft detected by Schrödinger SiteMap is highlighted in yellow. **b** Charge distribution in the reduced (PDB: 5ETC) and oxidized forms of p38α. We have added a layer of opacity to these representations while retaining the colors in the region of interest to highlight the differences in the docking cavity. **c** Close-up of the cleft (left) and the canonical binding site (right). Both binding sites are rotated 90° with respect to one another and are oriented as in I (reduced) and III, (oxidized) of Fig. 3a. In the oxidized form, the disulfide bond is located behind Q120 (not visible in III) as shown in Supplementary Fig. 3. Residues defining each binding pocket are colored by amino acid type and are identified by a single letter code, except for hydrophobic residues, that are listed below the panel. Conserved residues in both reduced and oxidized forms are shown in bold.

identified molecules so far had limited success as therapeutic treatments[2,29]. The existence of the oxidized form, which contains an intramolecular disulfide bond that has not been previously considered, may explain the difficulty in identifying functional compounds that bind to allosteric sites to regulate the function of p38α under oxidative stress. Furthermore, the redox state of the protein should be considered to increase the likelihood of identifying compounds that recognize regions other than the ATP binding site during the drug discovery process. Both p38α phosphorylation and oxidation can be measured by specific biochemical assays to test p38α functionality[13].

Our results suggest that the transient cavity created by the αD/LD redox-dependent conformational transition represents a target worth exploring in the future. A comprehensive search for candidates that target this transient cavity could yield promising compounds to stabilize the active or inactive conformations, even in the absence of the disulfide bond, opening new avenues for targeting p38α in disease.

## Methods

### Protein expression and purification

The mouse p38α cDNA (Uniprot: P47811) was amplified by PCR using the primers: CATCGCGAACAGATCGGTGGTGGTATGTCGCAGGAGAG GCCC and GTTTAAATGGTCTAGAAAGCTTCAGGACTCCATTTCTTC (Merck KGaA). and cloned into the pOPINS vector using an 'In-Fusion Cloning strategy'. The cloning was confirmed by DNA sequencing (GATC Biotech). Bacterial cultures (*E.coli* BL21) were induced at 37 °C (OD$_{600}$ of 0.7) and the protein was expressed at 20 °C for 6 h. Cultures were centrifuged and cells were lysed at 4 °C (EmulsiFlex-C5, Avestin) in 50 mM Tris, 500 mM NaCl, 10 mM imidazole and Tween 20 0.2% V/V pH 8 at 25 °C in the presence of lysozyme and DNase I (reagents purchased from Merck KGaA). The efficacy of this protocol has been previously described[21,30–32]. Supernatants containing the soluble proteins were loaded into a HiTrap™ His-Tag (Fisher Scientific) affinity column and eluted by an imidazole gradient to remove non-specifically bound bacterial proteins, using a NGC™ Quest 10 Plus Chromatography System (Bio-Rad). Fractions containing the protein were

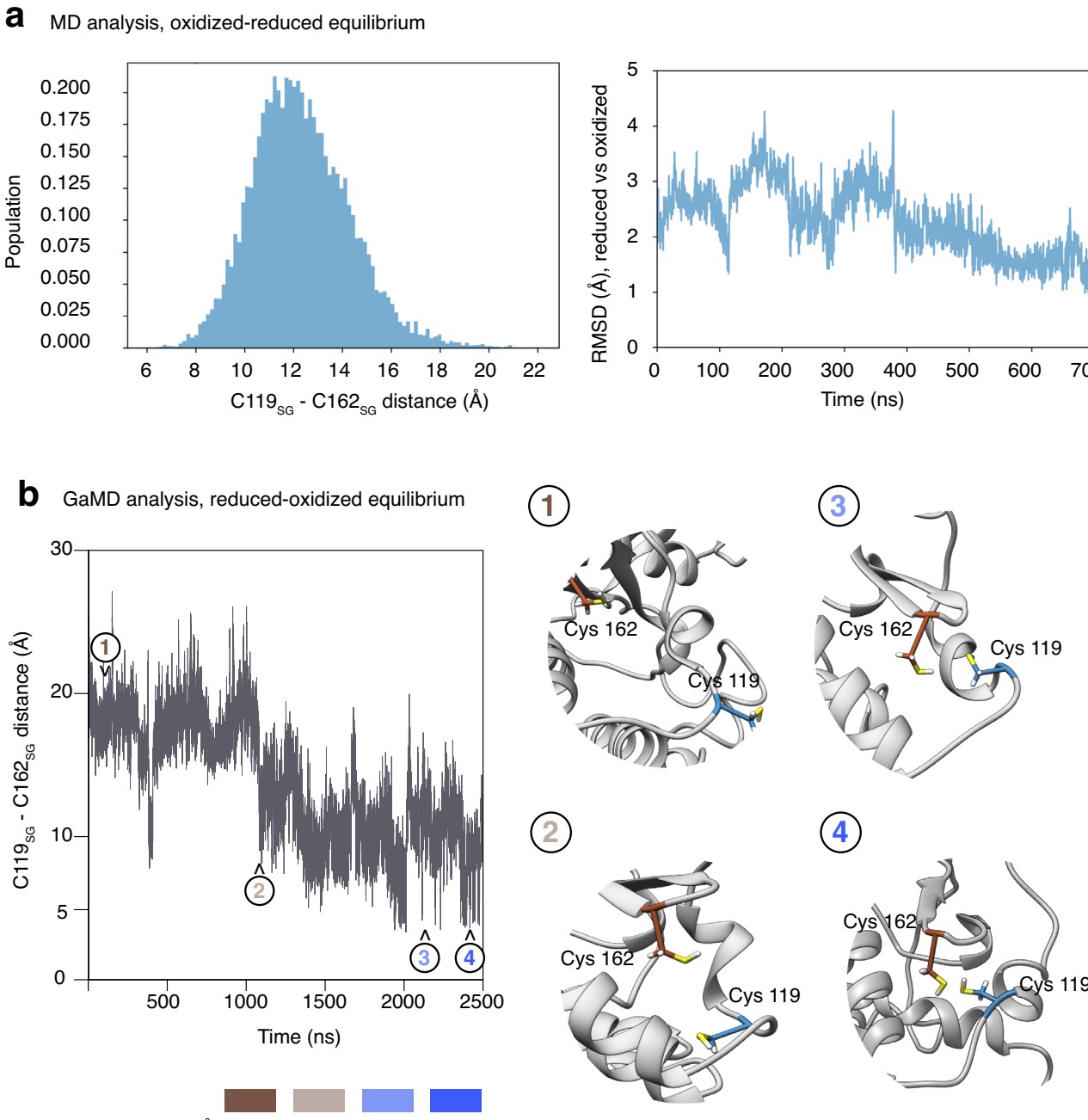

**Fig. 4 | Interconversion of oxidized and reduced p38α states. a** Molecular dynamics (MD) simulations were performed on the oxidized structures with the disulfide bond open, but starting with Cys119 and Cys162 side by side. Left: Histogram analysis of the distances between Cys119 and Cys162 along the MD simulations showing that the majority are in the range of 10–13 Å, which recapitulates the distances observed in the reduced structures captured in crystals (Fig. 1c). Right: RMSD values along the MD simulation of the 8ACO structure. Values are relative to the reduced 5R8W structure. At the end of the simulation, the RMSD is close to 1 Å,

indicating that after the release of the disulfide bond, the oxidized structure evolves towards the same fold observed in the crystals of the reduced form. **b** Gaussian accelerated MD simulation of the reduced p38α structure. Starting from the Cys residues separated by 18 Å, the simulation ends in folded conformations with both Cys side chains as close as 3.1 Å, capable of forming a disulfide bond. The four frames in the right panels illustrate the conformations explored during the simulation. The RMSD evolutions for the two simulations are shown in Supplementary Fig. 4.

pooled, dialyzed to reduce the NaCl concentration, and cleaved with recombinant Ulp1 (SUMO protease) overnight at 4 °C. Cleaved protein was loaded onto a HiTrap™ SP HP 5 mL (Fisher Scientific) column to separate the SUMO tag from p38α (N His SUMO Tag: 12.4 kDa/pI: 5.71 and p38α 41.3 kDa/pI: 5.55 and purified further by size-exclusion chromatography on a HiLoad™ 16/600 Superdex™ 75 pg (GE Healthcare Life Science) in 50 mM Tris, 150 mM NaCl and 1 mM DTT (For-Medium Ltd.) at pH 7.4. Aliquots were kept frozen at −80 °C.

### Crystallography
The diluted protein solution (100 μM) was dialyzed to remove the 1 mM DTT and left at 4 °C under air oxidation overnight, and full-length p38α protein (10 mg/mL, MW: 41.3 kDa) was crystallized in the presence of 1 mM SB203580 (Merck KGaA) in a solution containing 20 mM Tris pH 7.5, 100 mM NaCl, 10 mM MgCl₂ (VWR International Eurolab, S.L.). Screenings and optimization plates were prepared with total drop volumes of 200 nL each (1:1 protein/condition ratio)

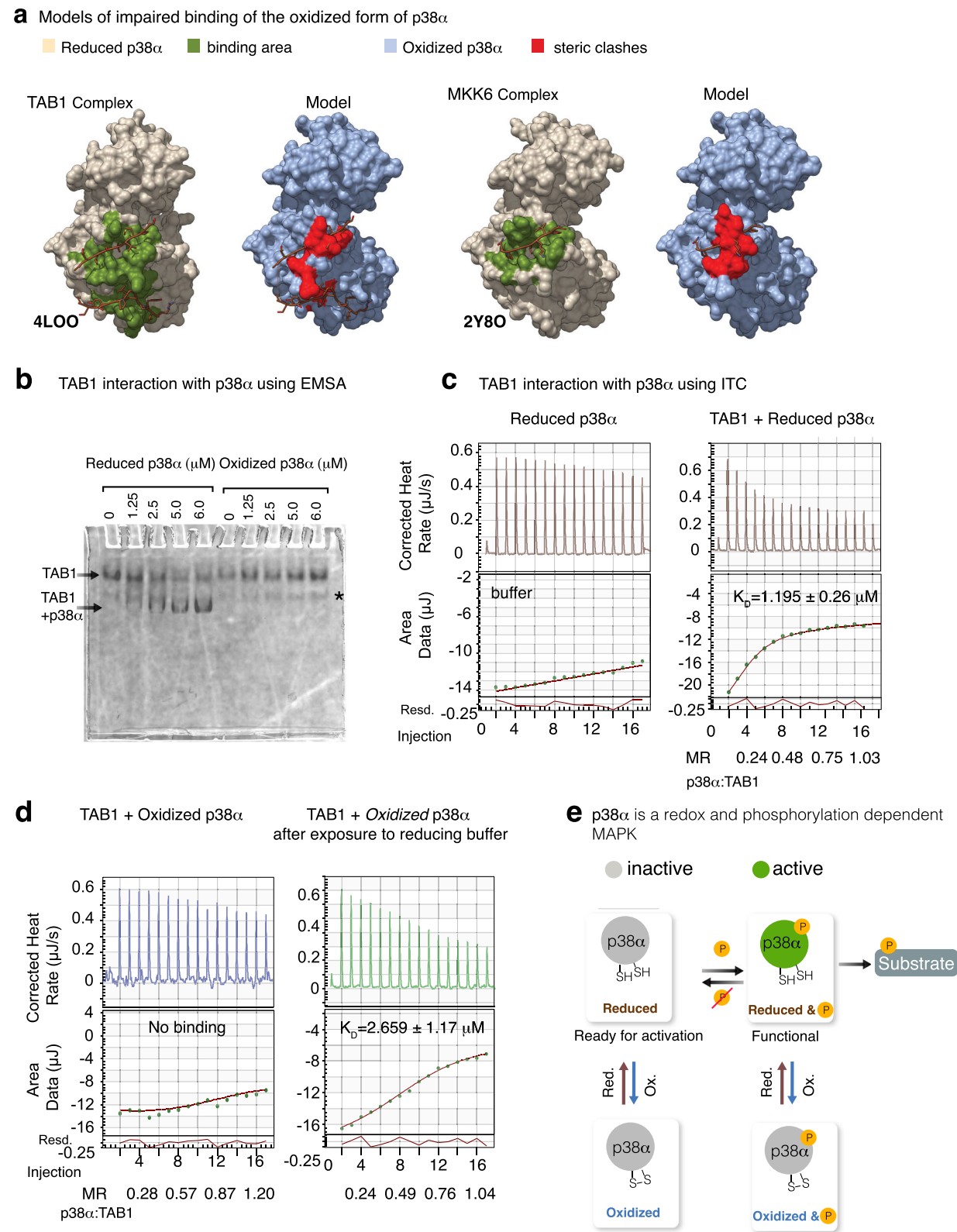

**a** Models of impaired binding of the oxidized form of p38α

Reduced p38α · binding area · Oxidized p38α · steric clashes

TAB1 Complex · Model · MKK6 Complex · Model

4LOO · 2Y8O

**b** TAB1 interaction with p38α using EMSA

**c** TAB1 interaction with p38α using ITC

Reduced p38α · TAB1 + Reduced p38α

$K_D=1.195 \pm 0.26$ μM

**d** TAB1 + Oxidized p38α · TAB1 + *Oxidized* p38α after exposure to reducing buffer

No binding · $K_D=2.659 \pm 1.17$ μM

**e** p38α is a redox and phosphorylation dependent MAPK

inactive · active

using an Art Robbins Phoenix Protein Crystallography Dispenser at the Automated Crystallography Platform (PAC) of IBMB-CSIC and IRB Barcelona. Crystals were grown by sitting-drop vapor diffusion and monitored on a Bruker Crystal Farm. Crystal growth conditions and data collection details are listed in Table 2. Diffraction data used for the structure determination were recorded at the BL13-XALOC beamline at the ALBA Synchrotron (Barcelona, Spain) with the

collaboration of ALBA staff[33]. Data reduction and processing was carried out using iMosflm 7.3[34] scala 3.3[35] and truncate from the CCP4 v8.0 suite[36]. Structures were solved by molecular replacement with Phaser 2.7[37,38], using the PDB structure 4LOO as the model reference. Refinement was carried out using REFMAC5 v5.5[39] and manually with Coot v.0.9.4.1[40]. UCSF Chimera v1.13.1[41] was used to prepare figures and calculate RMSD values for structural comparisons. Sequence

**Fig. 5 | Effects of the closed conformation of p38α on its interactions.**
**a** Representation of the TAB1 peptide (brown) bound to the reduced (left, PDB 4LOO, beige) and oxidized (right, model, blue) forms of p38α. The structures corresponding to MKK6-p38α (PDB 2Y8O) are also shown for comparison. As with TAB1, the complex is shown in beige and the model in blue. In both cases, most of the residues that form the cavity for the interaction in the reduced form (highlighted in green) have changed their position in the oxidized model (red), thereby precluding the interaction. Supplementary Movie 2 and Supplementary Movie 3 illustrate how the oxidized form interferes with ligand binding. **b** Oxidized and reduced p38α proteins were incubated at the indicated concentrations with the fluorescently labeled TAB1 peptide (0.25 μM in buffer with BSA 8 μg/mL). The complex formation was followed by Electrophoretic mobility shift assay (EMSA) analysis. The asterisk indicates a non-specific interaction with BSA. Two

independent experiments were performed (Supplementary Fig. 5b). **c** The reduced p38α protein was incubated either with buffer or with the fluorescently labeled TAB1 peptide and then analyzed by Isothermal titration calorimetry (ITC). Raw data is presented in the top panels and curve fitting of the data in the bottom panels, with all but the first titration point being used for the curve fitting. Residuals are displayed below the ITC plot. The apparent dissociation constant ($K_D$) and the Molar ratio (MR) are indicated. **d** ITC analyses were performed as in **c**, but using the oxidized p38α protein (left) or the oxidized protein after dialysis with the reducing agent TCEP in the buffer (right), which indicates the reversibility of the redox switch. **e** Proposed dual mechanism of p38α regulation, in which the functional kinase requires simultaneous reduction and phosphorylation in the activation loop for activity. Source data for this figure are provided as a Source Data file.

alignments were generated with the EMBL-EBI online server TCoffee v11 (https://tcoffee.crg.eu/).

### Electrophoretic mobility shift assay (EMSA)
To investigate the binding of TAB1 to p38α (qualitatively), we first dialyzed the p38α protein solution (200 μL of 20 μM) at 4 °C to remove the 1 mM DTT used for protein storage. Then, half of the sample was incubated with 10 μM $H_2O_2$ for 1.5 h to speed up the oxidation process, and the other half was incubated with 2 mM DTT to keep the protein in the reduced state. For the binding reactions, we mixed the FITC-TAB1 peptide (0.25 μM, amino acids 384-412 of TAB1, [RVYPVSVPYS-SAQSTSKTSVTLSLVMPSQ], GenScript) with BSA (8 μg/mL, Panreac) to prevent non-specific peptide binding, and incubated the mixture for 15 min on ice. Then we added increasing amounts of p38α (0–6 μM) in 20 μL of binding buffer (20 mM Tris pH 7.5, 100 mM NaCl and 20% DMSO, Panreac) with and without 2 mM DTT for the reduced and oxidized conditions, respectively. After further incubation for 15 min on ice, the reactions were stopped by adding 20 μL of Orange G Loading Dye 2X (Merck KGaA). Samples (10 μL) were loaded on a mini-PROTEAN® TGX™ precast gel (7.5% acrylamide, 10-well comb; Bio-Rad). This procedure was independently repeated twice. Gels were run in Tris/glycine running buffer pH 9.9 at 100 V for 1 h at 4 °C. Fluorescent images of the gels were acquired using a Typhoon™ 8600 (Cytiva). Experiments were performed in duplicates.

### Differential scanning fluorimetry (DSF)
The biophysical properties of the oxidized and reduced proteins were examined using DFS, performed in a real-time polymerase chain reaction (RT-PCR) system with SYPRO Orange fluorescent dye and analyzed using the HTS explorer as described[12]. Experiments were performed in duplicates.

### Isothermal titration calorimetry (ITC)
We followed the binding of TAB1 to p38α (quantitatively), using a nano ITC calorimeter (TA Instruments). Protein and peptide samples were dissolved in the same buffer and degassed before the experiments performed at 20 °C. Concentrations were determined using a Nano-Drop (ThermoFisher) system and their predicted extinction coefficients. The NanoAnalyze v3.7.5 software (TA Instruments) was used to analyze the binding isotherms. The cell (190 μL) contained 16 μM of TAB1 peptide in buffer 20 mM Tris pH 7.5, 100 mM NaCl. The syringe (50 μL) contained p38α protein at 60 μM in the same buffer. For the experiments with reduced protein, the protein buffer also contained 1 mM TCEP (ABCR GmbH & Co. KG) instead of DTT because the latter affects the stability of the baseline, presumably due to its rapid oxidation.

### Activity recovery dialysis
300 μL of oxidized p38α protein at 60 μM in 20 mM Tris pH 7.5, 100 mM NaCl, 10 mM $MgCl_2$ were dialyzed against 1 L of the same buffer with 1 mM TCEP, overnight at 4 °C. Afterwards, the protein was

centrifuged and quantified using a NanoDrop system, and concentrated to 60 μM. Aliquots were kept frozen at −80 °C.

### Molecular dynamics (MD)
The oxidized p38α structure (8ACO) was used as the starting model for MD, with the disulfide bond open, but with Cys119 and Cys162 as close as in the oxidized form. Simulations were performed with the Amber18 software package[42]. The protein was immersed in a pre-equilibrated octahedral water box with a 12 Å buffer of TIP3P water molecules using the Amber leap module, resulting in the addition of approximately 16 000 solvent molecules per system. The system was subsequently neutralized by the addition of explicit counterions ($Na^+$ and $Cl^-$). All calculations were done using the ff14SB Amber protein force field. A two-stage geometry optimization approach was applied, consisting of an initial minimization of solvent molecules and ions imposing protein restraints of 500 kcal mol$^{-1}$A$^{-2}$) followed by an unrestrained minimization of all atoms in the simulation cell. The system was then gently heated using six 50-ps steps, incrementing the temperature by 50 K in each step (0–300 K) under constant volume and periodic boundary conditions. Next, the system was equilibrated without restraints for 2 ns at a constant pressure of 1 atm and temperature of 300 K. Five MD production replicas of 700 ns each were performed in the NVT ensemble and periodic boundary conditions. For the Gaussian accelerated MD (GaMD) simulation of the transition between the reduced and oxidized forms of p38α, we used the ff99SB Amber force field on GPUs[43], starting from the reduced 5R8W structure. The conventional MD step was run for only 200 ns, followed by a 50 ns re-equilibration (1 atm, 300 K). The production GaMD was run for 2500 ns (Starting conditions for the two simulations are included in Supplementary Table 1).

### Binding site identification
The cleft was detected in the oxidized form by Schrödinger SiteMap 2021-2[44]. The solvent-accessible surface area for both the cleft and the canonical binding site (8ACM and 5R8W structures, without ligands) was calculated using fpocket 4.0[20].

### Reporting summary
Further information on research design is available in the Nature Portfolio Reporting Summary linked to this article.

## Data availability
The data that support this study are available from the corresponding authors upon request. The atomic coordinates of the oxidized structures have been deposited in the Protein Data Bank (PDB) with accession codes 8ACO and 8ACM. Datasets from PDB used in this study include: 5R8W, 4LOO, 5ETC, 3OBG, 1LEZ, 2Y8O, 1LEW, 3TG1, 2OZA. Additional entries used for preparation of Fig. 1b are detailed in Table 1. Movies have been prepared using 4LOO (TAB1) and 1LEZ (MKK3b).

The source data underlying Figs. 1d, 5c, d is provided as a Source Data file. Source data are provided with this paper.

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

## Acknowledgements

We thank the staff of the Automated Crystallography Platform (IBMB/CSIC- IRB Barcelona), the ALBA Synchrotron (BL13-XALOC beamline) and the Mass Spectrometry Core Facility (Universitat de Barcelona) for support, the Protein Expression Core Facility (IRB Barcelona) for providing reagents, and Dr. Susana de la Luna for insightful discussions. We would also like to thank Dr. Isabelle Brun-Heath for her help with protein purification and Laura Martínez-Castro for her input in setting up the GaMD simulations. B.B. was co-funded by the European Union's Horizon 2020 research and innovation programme under the Marie Skłodowska-Curie COFUND actions of IRB Barcelona and the PREBIST Predoctoral Programme, (agreement PREBIST_754558), L.G. was a recipient of a Severo Ochoa Predoctoral contract from MICINN (BES-2016-077122), R.A.M. is a recipient of a Joan Oró FI grant (AGAUR) and has been supported by a SOIB grant from the Servei d'Ocupació de Catalunya to promote access to science and technology for young researchers. Access to ALBA was granted through the BAG proposals 2018092972 and 2020094472. This work was financed by grants awarded to A.R.N. by the Spanish Ministerio de Ciencia e Innovación (MICINN, PID2019-109521RB-I00 and PID2022-136646OB-I00, funded by MCIN/AEI/10.13039/501100011033/ FEDER, EU), and the European Research Council (294665 and p38_InTh-825763), and by grants awarded by the Agency for Management of University and Research Grants (AGAUR) to M.J.M. (2021 SGR-866) and A.R.N. (2021 SRG-909) and from the BBVA foundation (M.J.M). We also acknowledge institutional funding from the CERCA Programme of the Government of Catalonia, IRB Barcelona, and MICINN through the Centres of Excellence Severo Ochoa award. M.O. is an ICREA Academia Fellow, and M.J.M and A.R.N. are ICREA Programme Investigators.

## Author contributions

Conceptualization was provided by M.J.M., J.P., M.O. and A.R.N., methodology by E.A., L.G., M.S., L.R., R.A.M., M.J.M. and J.P., investigations by E.A., B.B., L.G., R.A.M., L.R., J.P., J.I.F. P.M.M. and M.S., writing of the original draft by M.J.M., P.M.M. and J.P., writing of the final draft by M.J.M. and A.R.N. with input from all authors, visualization by P.M.M, J.P. and M.J.M., supervision by M.J.M. and A.R.N., funding acquisition by M.J.M. and A.R.N.

## Competing interests

The authors declare no competing interests.
