## [Peer Review File · Nature Communications]

Structural basis of a redox-dependent conformational switch that regulates the stress kinase p38 αReviewers' Comments:

Reviewer #1:

Remarks to the Author:

Review of "Structural basis of a redox-dependent conformational switch that regulates the stress kinase p38 α ". Pous et al. Review 20 Aug 2023.

This manuscript concerns the single crystal structure determination of an oxidized form of p38 α . The structure shows the formation of a disulfide bond and how it blocks the docking site in p38. The paper follows up on a 2010 paper showing that oxidation inhibits p38.

The structure presented clearly shows how oxidation is apparently inhibitory of p38 substrate interactions.

Figure 1 describes the conservation of cysteines across species. Fig 1B labels the p38 monomer (problems here). Fig 1C shows a wide distribution of distances between the two cysteines forming the disulfide bond, but that peptide complexes (substrates and activators) all have a similar distribution. Figure 2 shows that p38 is a monomer by gel filtration, that oxidation does not change the melting T_m, c) a schematic of the p38 structure d) the main result of the paper, the electron density for the disulfide, e) another schematic. Figure 3 is a ribbon diagram of the conformational change on oxidation, then three panels of low information content, a superposition of helix D conformers, then MD simulations and structures out of the MD simulations. Figure 4 is surface diagrams of the structure with a disordered helix D. Figure 5 is review material (docking interactions). Figure 6 is real data on pull down assays.

They do make an attempt to explain what the role might be of p38 inhibition in the presence of oxidative stress.

This paper is over-long and contains review material. A short paper on the primary data presented here, namely the structure of the disulfide bond in oxidized p38, would be welcome somewhere. The manuscript purports to be about structure, but lacks clear presentation of their own data. Where is the stick diagram showing the disulfide, and the electron density showing the disappearance of helix D? There are several labeling errors in Figures, especially Figure 1B (peptide binding groove wrong/poorly explained) and 2E (DGF?). Other Figures do not convey information well, especially Figure 2F and 3A,B. The molecular dynamics calculations coupled with distance measurements in Figure 3 is not helpful.

Reviewer #2:

Remarks to the Author:

This manuscript presents new insight into the regulation of p38 α , a Ser/Thr MAP kinase with critical roles in vital cellular processes, including stress responses. Although there are many reported structures of p38 α , this work is the first to define the structure and implications of a "mildly" oxidized state featuring a single Cys119-Cys162 disulfide.

The structural, computational, biochemical studies are carefully presented, and the work is of a high standard of rigor, and the topic is of broad interest and significance.

The authors show that the formation of the single new C119-C162 disulfide results in a significant conformational change that is likely biologically relevant under cellular conditions involving the accumulation of ROS species associated with stress. The oxidation is feasible in a stressed cellular environment; it is estimated that thiolate oxidation to disulfides in the correct ionization state can occur in living cells in the nM range of H₂O₂. Disulfide formation results in a movement of > 6.7 Å by Cys119 and triggers the unwinding of the α D helix. Most significantly, these changes also significantly

alter the p38 α docking site, which is essential for native kinase activity in healthy cells. The conformational change also reveals a new binding cleft. Together, these observations provide significant insight into the challenges with small molecule targeting of p38 α and the opportunities now that a new druggable site has been identified. Notably, the equivalent Cys pair also features in p38 β but is absent in other p38 paralogs and paralogs of JNK and ERK. Thus, the work also defines an opportunity for selective pharmacological targeting of an essential subset of the MAPKs.

In general, I find this manuscript highly suitable for publication in Nat Commun. I have some issues that should be addressed in a revision.

1. Citation 12 (PLoS) presents a method (PROP) for investigating the reversible oxidation in MAPKs including p38, and highlights the Cys pair in p38 α presented in this manuscript. I believe there should be a more detailed treatment of this foundational work when the citation is called out. The previous work does not in any way diminish the current contribution, but the method is important, and the findings speak to the greater role of such REDOX promoted structural changes and a method for screening them.

2. If possible it would be excellent if the authors could be more quantitative about conditions needed to promote oxidation. "Air oxidation" is somewhat qualitative. Starting with pure reduced protein under anaerobic conditions – can the authors define the minimum oxidizing conditions (e.g., a concentration of hydrogen peroxide) for Cys119-Cys162? This data type would connect the findings more closely to a cellular state that might induce the switch.

Minor

1. In the abstract – "Here we report an oxidized conformation..." should be: "...the conformation of an alternate oxidized state".

2. The use of "dormant" state does not seem quite right – perhaps "alternate"

Manuscript NCOMMS-23-34147-T.

Point-by-point response to reviewers

REVIEWER COMMENTS

Reviewer #1 (Remarks to the Author):

Review of “Structural basis of a redox-dependent conformational switch that regulates the stress kinase p38 α ”. Pous et al. Review 20 Aug 2023.

This manuscript concerns the single crystal structure determination of an oxidized form of p38 α . The structure shows the formation of a disulfide bond and how it blocks the docking site in p38. The paper follows up on a 2010 paper showing that oxidation inhibits p38.

We would like to add that we were not aware of the manuscript published in 2010 at the time we started our work. As mentioned in the manuscript, we found that the region around Cys119 is quite flexible, as evidenced by the number of different conformations captured in the structures deposited in the PDB. In some of them, Cys119 was relatively close to Cys162. All the complexes crystallized so far were obtained in the presence of reducing agents or using a Cys to Ser mutation. We thus decided to see if crystals grown in buffers without reducing agents could provide insights into how the p38 α structure behaves in non-reducing environments. At the same time, we wanted to learn if there were new conformations of this kinase that could be used as targets for drug discovery, and found that the presence of a disulfide bond alters the docking site region of p38 α . It was not until we solved the structures that we found the manuscript describing the inhibitory effects of p38 α oxidation in biochemical assays, which are compatible with our observations.

The structure presented clearly shows how oxidation is apparently inhibitory of p38 substrate interactions.

Figure 1 describes the conservation of cysteines across species. Fig 1B labels the p38 monomer (problems here). Fig 1C shows a wide distribution of distances between the two cysteines forming the disulfide bond, but that peptide complexes (substrates and activators) all have a similar distribution. Figure 2 shows that p38 is a monomer by gel filtration, that oxidation does not change the melting T, c) a schematic of the p38 structure d) the main result of the paper, the electron density for the disulfide, e) another schematic. Figure 3 is a ribbon diagram of the conformational change on oxidation, then three panels of low information content, a superposition of helix D conformers, then MD simulations and structures out of the MD simulations. Figure 4 is surface diagrams of the structure with a disordered helix D. Figure 5 is review material (docking interactions). Figure 6 is real data on pull down assays.

They do make an attempt to explain what the role might be of p38 inhibition in the presence of oxidative stress.

This paper is over-long and contains review material. A short paper on the primary data presented here, namely the structure of the disulfide bond in oxidized p38, would be welcome somewhere. The manuscript purports to be about structure, but lacks clear presentation of their own data. Where is the stick diagram showing the disulfide, and the electron density showing the disappearance of helix D?

Following the editorial recommendation, we have edited the manuscript to fulfill the guidelines of *Nature Communications*. We have also explained in more detail the structural findings, including new panels describing the disulfide bond present in both structures determined in this manuscript (new Fig. 2c and 2d).

Below we show the electron density for the region around both Cys residues, before and after refinement, to display that the electron density does not fit with the open and reduced structure, that the density for the helix D is absent (left) and that after building the disulfide bond, the structure fits the density well (right). The protein backbone is displayed to facilitate the recognition of the structural elements. Side chains for the Cys residues are also included. The structural differences caused by the formation of this bond and the absence of helix D are also displayed. This figure is included as a new Supplementary Figure 1 and is cited in the text (page 6).

There are several labeling errors in Figures, especially Figure 1B (peptide binding groove wrong/ poorly explained) and 2E (DGF?)

Thank you very much for pointing this out. We have corrected these mistakes.

Other Figures do not convey information well, especially Figure 2F and 3A,B. The molecular dynamics calculations coupled with distance measurements in Figure 3 is not helpful.

Regarding the molecular dynamics simulations, we would like to keep this information in the main text and figures, as it illustrates the reversible interconversion of the reduced and oxidized forms without inducing protein unfolding. These results are consistent with our binding assays as well as information available in the literature (PLoS 2010), which supports that both oxidized and reduced forms can interconvert depending on redox conditions both in vitro and in cellular assays. The simulations also captured the secondary structural changes that occur around Cys119 during the transition from the reduced to the oxidized state, allowing the formation of a disulfide bond.

To emphasize the importance of this section, we have now edited the description of these results (page 7) to help the reader understand how the reduced and oxidized forms can interconvert, and to emphasize that the disulfide bond does not represent a dead end in p38 α signaling. We have simplified the figure and reduced the number of snapshots shown. The selected ones focus on how the reduced and open conformations, in which the two Cys side chains are separated by up to 18Å, evolve to position them close enough to form the disulfide bond.

Please find below a List of modifications introduced in figures:

-The old panel 1b, describing the fold properties of p38 α is removed, and this information is incorporated in old panel 1c, now renamed as Fig. 1b. Old panels 2a and 2b describing the biophysical characterization of the oxidized and reduced forms of p38 α are now Fig.1d.

-The old panels 2c and 2e are now renamed as Fig. 2a and 2b respectively, and the information of old panel 2f (now removed) is described in Fig. 2b (bottom) and in the text (page 5), indicating that the inhibitor molecule binds the oxidized p38 α in the DFG-in conformation.

-The old panels 3a and 3b are edited, and renamed as Fig. 2e, to present in Figure 2 all the panels describing the main characteristics of the oxidized form. We have replaced the electron density panel with an identical one but with a white background.

-To streamline the story, we now present the molecular dynamics (MD) simulations after describing the new binding cavity created by the presence of the disulfide bond. Accordingly, we have renamed the old Figure 4 as Figure 3.

-Old panels 3c and 3d describing the MD simulations were edited, and are now Fig. 4a and 4b.

-The old Figure 5 is now mostly presented in supplementary Figure 1, except for a couple of panels that are now in Fig. 5a, which support the experimental data shown in this figure.

-The old Figure 6 is now presented as Figure 5b-5e.

Reviewer #2 (Remarks to the Author):

This manuscript presents new insight into the regulation of p38alpha, a Ser/Thr MAP kinase with critical roles in vital cellular processes, including stress responses. Although there are many reported structures of p38alpha, this work is the first to define the structure and implications of a “mildly” oxidized state featuring a single Cys119-Cys162 disulfide.

The structural, computational, biochemical studies are carefully presented, and the work is of a high standard of rigor, and the topic is of broad interest and significance.

The authors show that the formation of the single new C119-C162 disulfide results in a significant conformational change that is likely biologically relevant under cellular conditions involving the accumulation of ROS species associated with stress. The oxidation is feasible in a stressed cellular environment; it is estimated that thiolate oxidation to disulfides in the correct ionization state can occur in living cells in the nM range of H₂O₂. Disulfide formation results in a movement of > 6.7 Å by Cys119 and triggers the unwinding of the D helix. Most significantly, these changes also significantly alter the p38 docking site, which is essential for native kinase activity in healthy cells. The conformational change also reveals a new binding cleft. Together, these observations provide significant insight into the challenges with small molecule targeting of p38a and the opportunities now that a new druggable site has been identified. Notably, the equivalent Cys pair also features in p38beta but is absent in other p38 paralogs and paralogs of JNK and ERK. Thus, the work also defines an opportunity for selective pharmacological targeting of an essential subset of the MAPKs.

In general, I find this manuscript highly suitable for publication in Nat Commun. I have some issues that should be addressed in a revision.

1. Citation 12 (PLoS) presents a method (PROP) for investigating the reversible oxidation in MAPKs including p38, and highlights the Cys pair in p38a presented in this manuscript. I believe there should be a more detailed treatment of this foundational work when the citation is called out. The previous work does not in any way diminish the current contribution, but the method is important, and the findings speak to the greater role of such REDOX promoted structural changes and a method for screening them.

We agree with the reviewer that the work described in this citation is indeed complementary with our structural and binding results, as using different approaches we identify the same pair of Cys residues that modulate p38 α activity in a reversible and redox-dependent manner.

In the revised manuscript, we have expanded the text referring to this publication (pages 3, 9 and 10) and specifically mention the method for studying the reversible oxidation of p38 α in cells.

2. If possible it would be excellent if the authors could be more quantitative about conditions needed to promote oxidation. “Air oxidation” is somewhat qualitative. Starting with pure reduced protein under anaerobic conditions – can the authors define the minimum oxidizing conditions

(e.g., a concentration of hydrogen peroxide) for Cys119-Cys162? This data type would connect the findings more closely to a cellular state that might induce the switch.

In our experimental conditions for the structural studies, the oxidation reaction occurs spontaneously when the p38 α protein is exposed to atmospheric oxygen in the absence of any additional oxidizing agents, and the presence of reducing agents is required to maintain the protein reduced. We keep the stock protein solutions in the presence of 1 mM DTT to prevent aggregation.

For the p38 α and TAB1 binding reactions monitored by EMSA and ITC, we first dialyzed the p38 α protein solution (200 μ L at 20 μ M) to remove the 1 mM DTT present in the storage buffer. Our idea was to prepare the oxidized and reduced forms starting from the same p38 α solution and with comparable incubation times so that experimental conditions were as similar as possible. In this context, half of the solution was exposed to 10 μ M H₂O₂ (we have realised that the concentration was erroneously written as 10 mM in the previous version of the manuscript) for 1.5 h to speed up the oxidation process compared to the overnight process using air oxidation. The other half was incubated with 2 mM DTT for the same time to keep the protein in the reduced state. We have edited the Methods section of the revised manuscript to include this information.

We understand the comment of the referee, but since oxidation happens spontaneously, it makes it very difficult to estimate the minimal amount of H₂O₂ required for oxidation in vitro. Moreover, we believe that the information obtained from these experiments would be limited to the in vitro experimental conditions (under Argon atmosphere) and would be difficult to extrapolate to the cellular environment. We estimate that the spontaneous oxidation would likely equal to low nanomolar concentrations of H₂O₂ in vitro, but we did not examine these particular conditions in our protein crystallization screen or in the EMSAs.

In our opinion, the most relevant point is that we show that the oxidized and reduced p38 α states are interconvertible, and we describe experimental conditions that allow to prepare proteins in either of the two states in vitro to perform further functional characterizations.

Minor

1. In the abstract – “Here we report an oxidized conformation....” should be: “.....the conformation of an alternate oxidized state”.

We have edited the sentence as suggested.

2. The use of “dormant” state does not seem quite right – perhaps “alternate”

We have rephrased the two sentences to remove “dormant”.

Reviewers' Comments:

Reviewer #1:

Remarks to the Author:

Second review of "Structural basis of a redox-dependent conformational switch that regulates the stress kinase p38 α ". Pous et al. Review 24 Sep 2023.

In my initial review of this manuscript, I was negative about numerous mistakes in the figures, the lack of key figures, the inclusion of unhelpful calculations (molecular dynamics calculations and cavity finders), and the inclusion of review material. The revised manuscript is improved by including the appropriate figures, and the elimination of review material. Many labeling errors have been eliminated.

I still find the present paper thin for this journal. To bulk up the data on the structure of the disulfide bond blocking the docking site interactions in oxidized p38, the manuscript includes the idea that helix D of p38 adopts multiple conformations in different p38 crystal structures, molecular dynamics calculations, and protein stability measurements showing no change on oxidation, and cavity finder results showing a new cavity. The binding data in Figure 5 is interesting.

Specific remaining problems I observed:

Figure 1. Figure 1C needs an x-axis label that conveys the distance being plotted (Cys119S-Cys162S).

Figure 2. Figure 2A should be eliminated. This linear secondary structure information was published in the original descriptions of p38 α . There are numerous errors as presented, rendering this figure useless. Just to name a few, so the editor can understand, the ED site is not the P-Loop. The ED site is between beta 6 and 7; their structure does not have beta 8. There are inconsistencies between the labeling in 2A and 2B (APE?, alpha 2L12?).

The color scheme in the remaining panels should be defined. The rotation angle between the two diagrams in 2C should be stated.

Figure 3. Figure 3B adds little, since most of the electrostatic change is due to increased disorder in the A-loop. Figure 3C. Shouldn't the cysteines be indicated even if they are not on the surface?

Figure 4. Figure 4A adds very little.

Table 1. Shouldn't the oxidation state be listed as a heading with the PDB filename?

Reviewer #2:

Remarks to the Author:

The authors have carefully revised the manuscript to address all of the reviewer recommendations. I find this manuscript to be suitable for publication in Nat. Comm.

Answers (in blue) to reviewer#1 specific questions

Reviewer #1 (Remarks to the Author):

Second review of "Structural basis of a redox-dependent conformational switch that regulates the stress kinase p38 α ". Pous et al. Review 24 Sep 2023.

In my initial review of this manuscript, I was negative about numerous mistakes in the figures, the lack of key figures, the inclusion of unhelpful calculations (molecular dynamics calculations and cavity finders), and the inclusion of review material. The revised manuscript is improved by including the appropriate figures, and the elimination of review material. Many labeling errors have been eliminated.

I still find the present paper thin for this journal. To bulk up the data on the structure of the disulfide bond blocking the docking site interactions in oxidized p38, the manuscript includes the idea that helix D of p38 adopts multiple conformations in different p38 crystal structures, molecular dynamics calculations, and protein stability measurements showing no change on oxidation, and cavity finder results showing a new cavity. The binding data in Figure 5 is interesting.

Specific remaining problems I observed:

Figure 1. Figure 1C needs an x-axis label that conveys the distance being plotted (Cys119S-Cys162S).

We have modified the x-axis of Figure 1c to include this information.

Figure 2. Figure 2A should be eliminated. This linear secondary structure information was published in the original descriptions of p38a. There are numerous errors as presented, rendering this figure useless. Just to name a few, so the editor can understand, the ED site is not the P-Loop. The ED site is between beta 6 and 7; their structure does not have beta 8. There are inconsistencies between the labeling in 2A and 2B (APE?, alpha 2L12?).

We have removed the original panel 2a and renamed the remaining panels in the Figure 2 and in the text.

The color scheme in the remaining panels should be defined. The rotation angle between the two diagrams in 2C should be stated.

We have included an explicit description of the colors and the rotation value in Figure 2 and its legend.

Figure 3. Figure 3B adds little, since most of the electrostatic change is due to increased disorder in the A-loop. Figure 3C. Shouldn't the cysteines be indicated even if they are not on the surface?

We have described the differences in the classical docking site (reduced form) upon the formation of the disulfide bond (oxidized form described here). These differences include changes in both the surface shape and the electrostatic charge distribution of this particular region. To highlight the differences, we have added a layer of opacity to the structures of Figure 3b while retaining the colors in the region of interest, which does not contain the A-loop.

The disulfide bond is not visible in Figure 3c but the positions of Cys119 and Cys162 in the reduced form are shown. We now indicate the location of the disulfide bond in Figure 3a, and state in the figure legend that the disulfide bond is not visible in one of the orientations of the oxidized form presented. We have also added a new Supplementary Figure 3 to illustrate this point.

Figure 4. Figure 4A adds very little.

We respectfully disagree with the reviewer. Figure 4a shows that MD simulations starting from the conformation observed in the oxidized form can recapitulate what has been observed in crystals of the reduced forms (Figure 1c). We believe that the two simulations presented (Figures 4a and 4b) are complementary and highlight the reversibility of the redox switch, supporting the interconversion of the reduced and oxidized forms in the native context.

Table 1. Shouldn't the oxidation state be listed as a heading with the PDB filename?

We now indicate in Table 2 that the two structures determined correspond to the oxidized form.